# Studies of the Specific Activity of Aerosolized Isoniazid against Tuberculosis in a Mouse Model

**DOI:** 10.3390/antibiotics11111527

**Published:** 2022-11-01

**Authors:** Sergey V. Valiulin, Andrey A. Onischuk, Anatoly M. Baklanov, Sergey N. Dubtsov, Galina G. Dultseva, Sergey V. An’kov, Tatiana G. Tolstikova, Sergey N. Belogorodtsev, Yakov Sh. Schwartz

**Affiliations:** 1Voevodsky Institute of Chemical Kinetics & Combustion, Russian Academy of Sciences, 630090 Novosibirsk, Russia; 2N.N. Vorozhtsov Novosibirsk Institute of Organic Chemistry, Russian Academy of Sciences, 630090 Novosibirsk, Russia; 3Novosibirsk Tuberculosis Research Institute, Novosibirsk, 630040 Novosibirsk, Russia

**Keywords:** anti-tuberculous agent, isoniazid, inhalation, mice, histologic analysis, recovery rate

## Abstract

The aerosol inhalation delivery of isoniazid in mice was investigated, and the specific activity of the aerosol form of isoniazid was studied with the mouse model of tuberculosis infection, the *M. tuberculosis* H37Rv strain. Aerosol delivery was performed using a laminar-flow horizontal nucleation chamber. The inhalation dose was measured in real-time mode using a diffusion aerosol spectrometer. The mean particle diameter was 0.6 ± 0.03 μm, and the inhalation dose was 5–9 mg/kg. Pharmacokinetic measurements were carried out in nose-only and whole-body chambers. Isoniazid concentration in blood serum and its mass in the lungs were measured as a function of time using high-performance liquid chromatography. Studies of the specific activity of aerosolized isoniazid reveal that treatment with the aerosol lead to the complete recovery of the experimental tuberculosis infection as early as after 28 days after the start of inhalation treatment, while in the animals from the group receiving isoniazid per-orally, sole revivable tuberculosis mycobacteria were detected. Histologic examinations show that only a few macrophagal (nonspecific) granulomas without mycobacteria were detected in the spleen after per-oral and aerosol treatment, the number of granulomas on the 28th day being three times smaller in the latter case. The results show that the developed technique of isoniazid aerosol inhalation may have clinical potential.

## 1. Introduction

In spite of advances in the treatment of tuberculosis (TB) achieved during recent decades, this infection still remains an issue of great concern for public health facilities all over the world. Moreover, a substantial portion of medical efforts has been redirected to COVID-19 since the outburst of the pandemic, which has somewhat hindered the progress of treating TB. However, the involvement of the Earth’s population remains high: about one-third of the entire population is infected with latent TB infection, with one more case emerging every second [1]. Ten million people become new patients every year, and about 450 thousand of them suffer multidrug-resistant TB, and about four thousand people die every day from TB [2]. In order to reduce the development and spread of resistant forms of TB, a rapid diagnosis (for example, by whole-genome sequencing) and the correct setting of the treatment regimen, including the appropriate route of administration, are essential [3,4,5].

The treatment of drug-resistant TB is especially lengthy (up to two years) and involves second-line drugs that frequently cause severe side effects and complications [3]. In addition, some new anti-tuberculous drugs exhibit low bioavailability for per-oral administration, for example, due to poor solubility in water, so new routes of administration are to be elaborated [2]. Inhalation appears to be an efficient route to treat lung tuberculosis because, in this case, the lungs are the target organ for drug delivery [6].

The inhalation route for the treatment of TB has been extensively considered since the middle of the XX century. The first publications on aerosol inhalation therapy for tuberculosis appeared in print at the end of the 1940s [7,8,9,10,11]. During the years from 1950 to 1990, most of the works on pulmonary drug delivery in TB were published in the former Soviet Union. Those publications mostly considered a single-component aerosol delivery: streptomycin [12,13,14,15,16,17,18,19], ftivazide [18,20], kanamycin, florimycin [14,18], ethionamide [14], cycloserine [21], and rifampicin [22]. However, there were studies in which two or three anti-tuberculous substances were delivered by inhalation [23,24,25,26,27,28,29,30,31,32,33,34].

The evolution of multiple drug-resistant tuberculosis accelerated aerosol delivery investigations in the middle of the 1990s. The methods that were tested included delivery in the form of dry powder as well as nebulized drug forms [35]. Apart from the direct drug pulmonary delivery, the administration of anti-tuberculous drugs incorporated into excipient micron-sized particles has been under investigation during the past three decades. Many research groups studied the effect of drug incorporation into liposomes [36,37,38,39,40,41,42]. Porous and hollow nanoaggregates and microparticles were used as carriers [19,43,44,45,46,47,48,49]. Various agents have been developed for use as excipients that simplify the aerosol delivery of antituberculosis drugs. For instance, rifampicin has been administered through inhalation in association with silica [50], lipid carriers [51], and ethyl cellulose [52]. Improved lung delivery may be achieved using such excipients as metal-organic frameworks to overcome hepatotoxicity [53]. It has been demonstrated that the drugs incorporated into carrier particles can reduce dose frequency and improve the bioavailability of drug combinations [54]. However, this way of delivery is not without flaws. Storage problems arise from polymer destruction, particle agglomeration, settling, crystal growth, as well as drug leakage. The possible biological effects of all the studied excipients remain poorly understood yet. In addition, excipients need to be approved for respiratory drug administration [55]. Therefore, the elaboration of new excipient-free pulmonary drug delivery formulations is still necessary [56,57,58,59,60].

Notwithstanding some drawbacks of aerosol inhalation, this way of administration has some advantages, and therefore, various methods of pulmonary drug delivery are currently under development. Inhalation delivery to the lungs, as the primary site of TB infection, appears to be one of the robust potential strategies leading to improved efficiency, reduced doses and eliminated side effects. In contrast to injections, the aerosol route of delivery provides much higher drug concentrations locally at the site of infection. This is why the systemic level of drug exposure can be reduced with the conservation of therapeutic effect. To employ these advantages, it is necessary to elaborate on suitable delivery methods for antituberculosis drugs.

We have recently developed a procedure for the inhalation delivery of isoniazid, also known as Nydrazid, which is a hydrazide of isonicotinic acid, a drug that is recommended to treat TB of any localization [61]. Its side effects include headaches, dizziness, nausea, peripheral neuritis, and drug-induced hepatitis. We elaborated on an excipient-free procedure to generate the aerosol of this drug, within the particle size range of 80 nm–1.5 μm and developed instrumentation to deliver the aerosol to laboratory animals (mice) in nose-only and whole-body chambers [61,62,63,64,65,66,67]. The delivered dose was monitored with the help of a diffusion aerosol spectrometer, an instrument measuring particle size distribution and concentration. The pharmacokinetics of the drug was studied, along with its distribution over tissues. In the present work, we also assess the therapeutic effect of the aerosolized form of isoniazid for the murine model of TB, in comparison with per-oral ways of isoniazid administration. The goal of the present investigation is to reveal the specific activity of aerosolized isoniazid against tuberculosis in laboratory mice.

## 2. Experimental Procedure

### 2.1. Isoniazid Aerosol Generation and Inhalation Equipment

The aerosol generation system is described in detail elsewhere [61]. A laminar-flow horizontal evaporation-nucleation generator consisting of two chambers was employed. The first chamber is used to generate NaCl seeding particles of a mean diameter of 7 nm. It is made as a quartz tube with an inner diameter of 1.0 cm with an outer heater. The filtered air is supplied to the inlet of the tube with a flow rate of 0.8 L/min (at atmospheric pressure and room temperature). A quartz crucible with NaCl substance is inserted inside to be heated to 820 K. As a result, the air passing through is saturated with the NaCl vapor. At the outlet of the first chamber, the hot flow with sodium chloride vapor is mixed with cold air. The latter is supplied with a flow rate of 0.8 L/min. As the temperature decreases, the vapor becomes supersaturated, and homogeneous nucleation starts, resulting in the formation of NaCl particles. The flow with NaCl particles is supplied to the second chamber, which is a quartz tube with an inner diameter of 2.5 cm, heated to 450 K with an outer oven. A crucible with isoniazid is inserted into the hot zone of the second chamber. As a result, the airflow with NaCl particles is saturated with the isoniazid vapor. As the temperature decreases at the outlet of the second chamber, heterogeneous nucleation starts on the seeding particles. The resulting aerosol of isoniazid is then mixed with pure air, which is supplied with a flow rate of 1.5 L/min. Then, the aerosol is directed to an inhalation chamber with the laboratory mice. The arithmetic mean diameter of the isoniazid aerosol used in this work was 0.6 ± 0.03 μm.

The pharmacokinetic studies were performed using both nose-only (NO) and whole-body (WB) inhalation chambers. The isoniazid-specific activity was studied using WB chambers. The WB chamber is a quartz cylinder 40 cm long, with an inner diameter of 9 cm, equipped with two end trapdoors made of stainless steel. The trapdoors are equipped with pipe fittings to let the aerosol flow in and out. The animals are free to move inside the chamber during inhalation. In the NO chamber, mice are placed radially in two tiers, six animals in each one, around the cylindrical aerosol compartment. Only the nose of a mouse is exposed to the aerosol. To control the dose delivered to the laboratory mice during the whole experiment, the concentration and size of the aerosol particles were monitored at the outlet of the inhalation chamber. To provide additional control to the aerosol mass concentration, samplings on the glass fiber aerosol filter Whatman GF/A (25 mm) were carried out. The mass of the deposit was determined by weighing it with the analytical balance.

Outbred laboratory male mice CD-1 were used for the pharmacokinetic experiments in this work. The animals were taken from the SPF vivarium of the Federal Research Center Institute of Cytology and Genetics of the Siberian Branch of the Russian Academy of Sciences. The investigation of specific activity was carried out with male BALB/c mice. The mice were obtained from the animal facility at the State Research Center of Virology and Biotechnology Vector. Both CD-1 and BALB/c mice were 4 months old, with a body mass of 22 ± 2 g. The study was approved by the Ethics Committee of Novosibirsk Tuberculosis Research Institute (Protocol No. 45/1 of 10.11.2019) and conducted in compliance with Directive 2010/63/EU The European Parliament and the EU Council for the Protection of Animals Used for Scientific Purposes.

The concentration and size distribution of the isoniazid aerosol particles are determined at the outlet of the reactor with the help of a DSA-M diffusion aerosol spectrometer [59,60,61,62,63] created at the Voevodsky Institute of Chemical Kinetics and Combustion, Novosibirsk, Russia. The device is able to measure the particle number concentration up to 5 × 10^5^ cm^−3^ (without preliminary dilution) and particle size distribution within a range of 3–1100 nm. The instrument operates on the basis of the recovery of the particle size distribution from the known dependence of particle mobility on their size. A non-selective aerosol diluter was used to decrease the particle concentration to fit the range recordable for DSA-M.

### 2.2. Inhalation Dose

In the experiments with NO chambers, the total dose delivered by inhalation can be measured directly. The total mass of isoniazid particles deposited in the respiratory tract of a mouse per time *t*_0_ (min) follows the formula:(1)$$\Delta M=\alpha C_{A}Ft_{0}$$
where *C_A_* (mg/cm^3^) is the particle mass concentration in the aerosol chamber, *F* (cm^3^/min) is the aerosol flow rate through the inhalation chamber, *α* is the mean fraction of particles that a mouse can consume by inhalation from the aerosol flow passing through the chamber. The quantity *α* is determined as:(2)$$\alpha =\frac{1}{N}\frac{n_{0}-n}{n_{0}}$$
where *N* is the number of mice in the chamber. The quantities *n* and *n*_0_ (cm^−3^) are the outlet aerosol number concentrations for a chamber occupied with mice and for a non-occupied one, respectively. Thus, the quantity $n_{0}-n$ gives the number of consumed particles per unit volume. The combination of Equations (1) and (2) gives the inhalation dose *D* (mg/kg) per unit weight of the mouse:(3)$$D=\frac{\Delta M}{m}=\left(1-\frac{n}{n_{0}}\right)\frac{C_{A}Ft_{0}}{Nm}$$
where *m* is the mean mass (in kg) of mice in the chamber.

The total mass of particles delivered to the respiratory system of laboratory mouse during the aerosol exposition can also be expressed as:(4)$$\Delta M=C_{A}\varepsilon v_{m}t_{0}$$
where *ε* is the total deposition efficiency in the respiratory ways, that is, the ratio of the number of particles from the inhaled volume deposited in the respiratory ways to the number of particles contained in this volume of air in the aerosol chamber, *v*_m_ is the minute volume, i.e., the total volume inhaled by a mouse per one minute. The quantity *v*_m_ (cm^3^/min) can be estimated for the experiments in the NO chambers as [68]:(5)$$v_{m}\approx \left(600\;\pm \;100\right){\left(m/kg\right)}^{0.75}$$

The value *ε* can be determined experimentally using the following expression obtained by a combination of Equations (1), (2), and (4):(6)$$\varepsilon =\left(1-\frac{n}{n_{0}}\right)\frac{F}{v_{m}Nm}$$

The efficiency of deposition in the respiratory ways is a function of the particle diameter *d*. Using Equation (6), we determined experimentally the deposition efficiency in the NO experiments with mice for the range of particle diameters at 10–2000 nm, which can be expressed as a sum of two Gaussian functions:(7)$$\varepsilon (\ln(d))=0.85\exp\left(-\frac{1}{2}{\left(\frac{\ln\left(d/4.0(nm)\right)}{2.2}\right)}^{2}\right)+0.60\exp\left(-\frac{1}{2}{\left(\frac{\ln\left(d/1590(nm)\right)}{1.1}\right)}^{2}\right)$$

In pharmacokinetic experiments with NO chambers, the dose delivered to the respiratory ways was determined as:(8)$$D\left(\mathrm{mg}/\mathrm{kg}\right)=\frac{\Delta M}{m/\mathrm{kg}}=\frac{C_{A}\varepsilon v_{m}t_{0}}{m/\mathrm{kg}}$$
using the particle deposition efficiency as determined by Equation (7) and *v_m_* calculated according to Equation (5).

One should note that the mouse minute volume is a function of sex, strain, age, state of health, and other parameters. Therefore, there is some error in determining the minute volume by the formula Equation (5). However, the accuracy of inhalation dose acquisition by formula Equation (8) is independent of uncertainty in the minute volume because the quantity *ε* is inversely proportional to *v_m_*, and the dose depends on the product *εv_m_*.

The minute volume for animals in the WB chambers is higher than that for animals immobilized in NO chambers by about 30% [69,70,71,72,73,74]. Therefore, Equation (5) can be rewritten for the mice in WB chambers as:(9)$$v_{m}\approx \left(800\;\pm \;100\right){\left(m/kg\right)}^{0.75}$$

Thus, the inhalation dose was calculated in this work by Equation (8), with the minute volume determined by Equations (5) and (9) for the mice in the NO and WB chambers, respectively.

### 2.3. Sample Preparation and Chromatographic Analysis

The isoniazid content in the serum and organs was determined by means of high-performance liquid chromatography (HPLC) in the ion-pair version. HPLC was chosen due to its high sensitivity, selectivity, the possibility to determine several substances in a single chromatogram, and applicability for serial analyses. Sample preparation was carried out as follows.

To determine isoniazid concentrations in the murine blood serum, trifluoroacetic acid (TFA) (Fisher Chemical, 99+%, for HPLC) 50% was added to serum samples at a ratio of 4:1 (V_sample_: V_TFA_), the mixture was intensively stirred for 5 min with Multi-Vortex V-32, centrifuged for 15 min with CM-50 centrifuge at 15,000 r.p.m., and then the supernatant was sampled (V = 100 μL) into chromatographic tubes.

The lungs and liver samples were cut and treated in the ultrasonic homogenizer after adding 200 μL of water. The homogenate was transferred into a plastic tube, TFA 50% was added at a ratio of 1:4 (V_TFA_: V_sample_), stirring was carried out for 5 min, centrifuging for 15 min at 15,000 r.p.m., and then 100 μL of each sample was transferred into chromatographic tubes.

Chromatographic analysis of isoniazid in the blood serum, liver, and lungs was carried out using a column filled with the reverse-phase sorbent ProntoSil 120-5-C18 AQ by means of a gradient elution: eluent A—acetonitrile (Sigma-Aldrich, for HPLC, gradient grade, ≥99.9%,), eluent B—water with heptyl sulfonate (Fisher Chemical, for Ion Pair Chromatography) (0.4 %) and TFA (0.1 %). Acetonitrile content in the eluent varied from 1 to 60 % within 2000 μL. The volume of a sample introduced into the chromatograph was 20 μL, the elution rate was 150 μL/min, and the column was thermostated at 40 °C. Detection was carried out with the UV absorption detector at wavelengths of 254, 266, 280, and 310 nm. Examples of the chromatograms of the intact blood serum, blood serum with isoniazid added, the intact lungs and lungs with isoniazid added are shown in Figure 1.

### 2.4. Specific Activity in the Model of Acute Tuberculosis Infection

#### 2.4.1. Tuberculosis Infection Model and Experimental Groups

When investigating the specific activity, tuberculosis infection was modeled with the *M. tuberculosis* (MBT), strain H37Rv, obtained from the collection of the Bacteriological Laboratory of the Novosibirsk Research Institute of Tuberculosis (NRIT), Ministry of Health RF. MBT was introduced intravenously into the retroorbital sinus in the amount of 2 × 10^7^ MBT/mouse in the buffered physiological solution. The animals were divided into groups, with 30 individuals in each group. Groups 1 and 2 are non-treated ones (Control (−) and Control (+)), Group 3 was treated with the per-oral form of isoniazid, and Groups 4 and 5 were treated by aerosol inhalation delivery (see Table 1).

On the 14th day after infection, 2 mice from each group were euthanized, and their organs were sampled for histological examination. Acid-fast mycobacteria (AFM) and specific tuberculous lesions with prevailing spleen affection were detected in the animals of all groups except for control Group 1.

After infection, the animals were kept in isolated cages with independent ventilation (Independent Ventilated Cages, IVC), which excluded contact between animal groups and cross-infecting. The animals were kept with free access to food and water. All manipulations with animals were carried out in agreement with the European Convention for the Protection of Vertebrate Animals Used for Experimental and Other Scientific Purposes (Strasbourg, 1998) and were approved by the Ethics Committee of the NRIT.

The animals were withdrawn from the experiment through neck vertebra dislocation on the 28th and 56th day after infection. The lungs, liver, and spleen were sampled for histologic and bacteriological examination.

#### 2.4.2. Histologic Examination

To evaluate the mycobacterial load, histologic sections of the lungs (the superior lobe of the left lung), liver (the middle right hepatic lobe), and spleen (the ventral half) were prepared, and Ziehl–Neelsen coloration was performed. No less than 5 sections 5 μm thick were made from each preparation, acid-resistant MBT was visualized in 60 visual fields from each section. The data are presented as the number of MBTs per visual field. The manifestation degree of the specific inflammatory process was evaluated after coloring with hematoxylin-eosine over 60 visual fields from each preparation. For the spleen and liver, data were expressed as the number of granulomas per visual field, taking into account only specific granulomas containing epithelioid cells. The sections were examined by two independent researchers.

#### 2.4.3. Bacteriological Examination

For bacteriological examination, the fragments of the lungs, liver, and spleen, after preliminary weighing, were homogenized using a mechanical method. About 10% of the obtained homogenate was used to prepare smears and to make coloration with auramine O rhodamine (HiMedia Laboratories, Mumbai, India). Fluorescently colored mycobacteria were counted using a luminescence microscope (Axio Lab A1 FL-LED, Carl Zeiss Microscopy GmbH, Jena, Germany) in 100 visual fields. In addition, parts of the homogenates of the same volume were plated onto a solid Löwenstein-Jensen medium (BioMedia, Saint Petersburg, Russia), incubated for 12 weeks at 37 °C, and then the number of colony-forming units (CFU) was counted.

#### 2.4.4. Statistical Processing

Statistical processing of the obtained data was carried out using STATISTICA 6 software; the normality of distribution was tested with the help of the Kolmogorov–Smirnov criterion. The data are presented in the work as the medians or as the mean values. To evaluate the reliability of differences between the experimental groups, the Student’s test was used (for the normal distribution) or Mann–Whitney test (for non-parametric distribution), and differences were considered reliable for the confidence interval at a level of 95% (*p* < 0.05).

## 3. Results and Discussion

### 3.1. Pharmacokinetics of Aerosolized Isoniazid

The average diameter of the aerosol particles of isoniazid involved in the experiments was 600 ± 30 nm, and the particle size of the spectrum was well described by the lognormal distribution with *σ_g_* = 1.4. The aerosol mean arithmetic concentration in the inhalation chambers was (4.5 ± 0.2) × 10^6^ cm^−3^, and the mass concentration was *C_A_* = 650 ± 50 mg/m^3^.

The isoniazid concentration in the blood serum as a function of inhalation time is presented in Figure 2a. Mice were exposed to isoniazid aerosols during a definite time interval (inhalation time). After the termination of the exposition, the mice were sacrificed immediately. The mass of isoniazid in the lungs came to saturation after about 30 min of inhalation (Figure 2b), which probably meant that at this time, the rate of isoniazid elimination from the lungs was about equal to the rate of lung delivery. The isoniazid concentration in the serum vs. time can be satisfactorily described in terms of the one-compartmental model. In this case, the kinetic equation is:(10)$$\frac{dC}{dt}=\frac{I}{V_{d}}-k_{e}C$$
where *C*(μg/cm^3^) is isoniazid concentration in the serum, *I* (μg/min) is the rate of aerosol delivery, *k_e_* is the elimination rate constant, *V_d_* (cm^3^) is the volume of distribution, i.e., the theoretical volume that would be necessary to contain the total amount of the administered drug at the same concentration as that observed in the blood plasma. The solution of Equation (10) is:(11)$$C=\frac{I}{V_{d}k_{e}}\left(1-exp\left(-k_{e}t\right)\right)$$

To fit the experimental points given in Figure 2a by Equation (11), the quantities $\frac{I}{V_{d}}$ and *k_e_* were adjusted. The best fit values were:(12)$$\frac{I}{V_{d}}=0.4\pm 0.02\;\mu g/(\min\cdot {\mathrm{cm}}^{3}),$$
*k_e_* = 0.019 ± 0.003 min^−1^ (solid line in Figure 2a). The rate of aerosol delivery can be evaluated as:(13)$$I=C_{A}v_{m}\varepsilon =0.65\;\mu g/{\mathrm{cm}}^{3}\cdot 34{\;\mathrm{cm}}^{3}/\min\cdot 0.47\approx 10\;\mu g/\min$$

Then, from Equations (12) and (13), we get *V_d_* = 25 ± 3 cm^3^ which is in a good agreement with the typical value of the volume of distribution *V_d_* = 23 cm^3^ as determined elsewhere [75]. The elimination rate constant *k_e_* = 0.019 ± 0.003 min^−1^, as determined from the data presented in Figure 2a is in a reasonable agreement with the value *k_e_* = 0.016 ± 0.010 obtained elsewhere [75,76,77,78,79].

The concentration of isoniazid in the serum and its mass in the lungs during and after inhalation is shown in Figure 3. Both the concentration of isoniazid in blood and its mass in the lungs increased with time during inhalation. After the termination of inhalation, the mass of isoniazid in the lungs started to decrease due to its absorption in blood circulation. However, the drug concentration in the serum started to decrease only about 20 min after the termination of inhalation. This delay in serum concentration decrease was because the mass of isoniazid in the respiratory system was still high at the inhalation stop time, and lung-to-blood absorption was still concurrent with the drug elimination from the blood.

To determine the bioavailability of aerosol delivery, the isoniazid concentration vs. time was measured for intravenous (IV) administration (Figure 4a). The elimination rate constant was determined from these data to be *k_e_* = 0.019 min^−1^, which is in agreement with the value obtained from aerosol delivery experiments.

An important value providing evidence of the efficiency of the drug penetration into the blood is the area under the curve (AUC) of the dependence of concentration on time from the start of introduction until excretion from the organism. Thus, for the data presented in Figure 3a, AUC_A_ for aerosol introduction is 510 ± 50 (μg min)/cm^3^, and for IV introduction, it is AUC_A_ = 300 ± 40 (μg min)/cm^3^. Taking into account the doses D_A_ = 9.0 ± 1.0 mg/kg and D_IV_ = 5.0 ± 0.5 mg/kg = 110 μg delivered in the aerosol and IV forms, respectively, the bioavailability *F* of the aerosol form of isoniazid can be calculated as:(14)$$F=\frac{AUC_{A}D_{IV}}{D_{A}AUC_{IV}}100\%=\frac{510*5.0}{300*9.0}100\%=(94\pm 10)\%$$

This means that the bioavailability of aerosol forms of isoniazid are close to 100%.

The volume of distribution can be determined from the IV delivery experiments as:(15)$$V_{d}=\frac{D_{IV}}{k_{e}AUC_{IV}}=\frac{110}{0.019*300}=19\;\pm \;3{\;\mathrm{cm}}^{3}$$
which is in a reasonable agreement with that obtained for aerosol delivery.

It is of interest to compare the temporal dependence of isoniazid concentration in serums resulting from per-oral delivery (Figure 4b) with that for the aerosol ways of administration. The rate of absorption from the gastrointestinal (*GI*) region to the blood (*W_GI_*) can be approximated by the first-order kinetics:(16)$$W_{GI}\left(t\right)=-\frac{dC_{GI}}{dt}=k_{GI}C_{GI}$$
where *k_GI_* is the first-order rate constant for isoniazid transfer through the *GI* barrier, *C_GI_* is the equivalent mass concentration of isoniazid in the *GI* tract, and *t* (min) is time. Then, the kinetic equation for the isoniazid concentration *C*(*t*) in serum can be written as:(17)$$dC\left(t\right)/dt=k_{GI}C_{GI}-k_{e}C\left(t\right)$$

The joint solution of Equations (16) and (17) is:(18)$$C\left(t\right)=\frac{k_{GI}C_{GI}^{0}}{\left(k_{e}-k_{GI}\right)}\left(\exp\left(-k_{GI}t\right)-\exp\left(-(k_{e}t\right)\right)$$
where $C_{GI}^{0}$ is the initial equivalent mass concentration of isoniazid in the *GI* region. To fit the experimental points presented in Figure 4b by Equation (18), three parameters *k_GI_*, *k_e_*, and $C_{GI}^{0}$ are to be adjusted. The best fit values are *k_GI_* = 0.16 ± 0.005 min^−1^, *k_e_* = 0.019 ± 0.005 min^−1^, and $C_{GI}^{0}$ = 4.3 ± 4 μg/cm^3^. It is important to note that the quantity *k_e_* determined from the data on the per-oral administration is in a good agreement with those determined from the IV and aerosol delivery (Figure 4a). The volume of distribution determined from the per-oral delivery experiments is:(19)$$V_{d}=\frac{D_{GI}}{k_{e}AUC_{GI}}=\frac{110}{0.019*230}=25\;\pm \;3{\;\mathrm{cm}}^{3}$$
where *D_GI_* = 5 mg/kg = 110 μg is the per-oral dose. The volume of distribution determined for per-oral administration is in a good agreement with those obtained from both the aerosol and IV delivery experiments.

### 3.2. Specific Activity in the Model of Acute Tuberculosis Infection

The overall condition, external view, mobility, and attitude to food were satisfactory and did not differ in the experimental or reference groups. The body mass of the experimental and reference animals insignificantly increased during the experiment in all groups, and there was no significant difference between the groups.

The only macroscopic finding was splenomegaly in the animals from the Control (+) group; the highest extent was detected on the 28th day (Figure 5). In all other groups, spleen mass did not differ from that in the intact mice.

The histologic investigation of non-infected animals did not reveal any pathological changes. During the histologic examination on the 28th day, weakly pronounced nonspecific interstitial lymphocytic-macrophagal infiltration was observed for the infected Control (+) group, with the thickened interalveolar septum (Figure 6a); on the 56th day, the manifestation of the inflammatory process was conserved at the same level. Specific epithelioid cells and tuberculosis mycobacteria were not detected.

In the liver, rare nonspecific inflammatory accumulations of mononuclear cells were detected on the 28th and 56th days; no specific granulomas or *Mycobacterium tuberculosis* (MBT) were detected (Figure 6b).

In the spleen, specific tubercular granulomas, in particular confluent ones, were detected only in animals of the Control (+) group. These granulomas always contained MBT, sometimes in large amounts (Figure 6c,d). Lymphoid-macrophagal hyperplasia and congestion were detected. Spleen preparations of the animals from other experimental groups contained only sole small macrophagal (nonspecific) granulomas without MBT. The average number of granulomas per visual field and the MBT detected in the murine spleen on the 28th and 56th days are presented in Table 2.

Attention should be paid to the fact that the number of granulomas in the spleen of mice from the Aerosol (1) and Aerosol (2) groups on the 28th day was about three times smaller than in the spleen of mice from the Per-oral group.

The results of the bacteriological investigation confirmed the histologic data. On the 28th day, acid-fast mycobacteria (AFM) occurred in large amounts in the spleen (312 ± 56.3) of animals from the Control (+) group, but they were rare in the liver and in the lungs. In the organs of animals treated with isoniazid, the sole AFM was found only in the liver and lungs after the per-oral introduction of the drug. In both groups treated with isoniazid in the form of nanoaerosol, AFM was not found.

On the 56th day, the most heavily affected organs in the animals of the Control (+) groups were the lungs (152 ± 46.2 AFM per 100 visual fields) and spleen (81.2 ± 33.8 AFM per 100 visual fields). No AFM was detected in all the groups treated with the drug under investigation, independently of the route of delivery. The average number of AFM per 100 visual fields, calculated for all the organs (lungs, liver, spleen), is presented in Table 3.

After inoculation on a Löwenstein–Jensen medium, on the 28th day, the average number of colony-forming units (CFU) in the Control (+) group was 77.2 ± 65.5, as in the previous case, and the spleen was the most strongly affected organ (70.2 ± 64.1 CFU per mouse). In the group of animals treated with isoniazid per-orally, the sole CFU was observed after the inoculation of spleen homogenate. In the Aerosol (1) and Aerosol (2) groups, no CFU growth was detected in all the organs. On the 56th day, the spleen was still the most strongly affected organ in the Control (+) group (11.4 ± 8.6 CFU per organ), but lungs were also observed to be affected (7.4 ± 7.1 CFU per organ), as well as the liver (3.8 ± 2.8 CFU per organ). No CFU growth was detected in all the groups treated with the drug under investigation, independently of the administration method (Table 3). These data confirm the efficiency of aerosolized isoniazid at a level not lower than that of the per-oral form, though the lower number of nonspecific granulomas in the spleen of aerosol-treated mice in comparison with that treated per-orally and the absence of AFM in the organs of aerosol-treated mice (against sole AFM in the liver and lungs of the mice treated per-orally) suggest even higher efficiency of the aerosolized form of isoniazid.

## 4. Conclusions

The inhalation delivery of isoniazid was investigated in comparison with the per-oral method. For this purpose, the inhalation system composed of a two-section evaporation-nucleation aerosol generator and a whole-body/nose-only inhalation chamber was employed. The delivered dose was monitored in real-time mode with the help of a diffusion aerosol spectrometer DSA-M, and the software was developed especially for the instrument. The size of the generated aerosol particles is well described by the lognormal distribution with an average size of 600 ± 30 nm and *σ*_g_ = 1.4.

Pharmacokinetic studies show that the bioavailability of isoniazid after inhalation delivery is close to 100 %.

The specific activity of the aerosol form of isoniazid was studied with the model of tuberculosis infection and *M. tuberculosis* H37Rv strain. It is demonstrated that the application of isoniazid in the form of aerosol in doses between 5 and 9 mg/kg leads to the complete recovery of the experimental tuberculosis infection as early as 28 days after the start of inhalation treatment, while in the animals from the group receiving isoniazid per-orally in the dose of 10 mg/kg, sole revivable tuberculosis mycobacteria were detected.

## Figures and Tables

**Figure 1 antibiotics-11-01527-f001:**
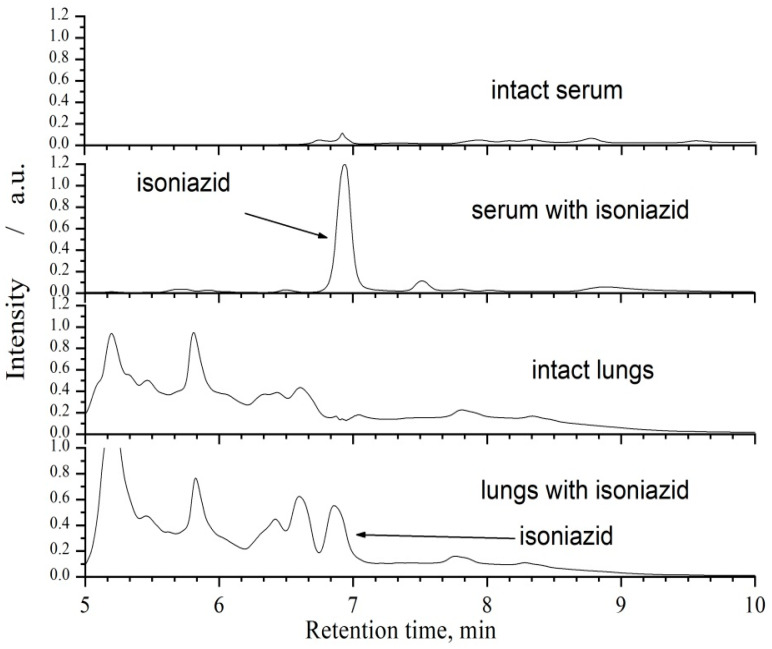
Examples of chromatograms of intact blood serum, blood serum with isoniazid added, intact lungs and lungs with isoniazid added.

**Figure 2 antibiotics-11-01527-f002:**
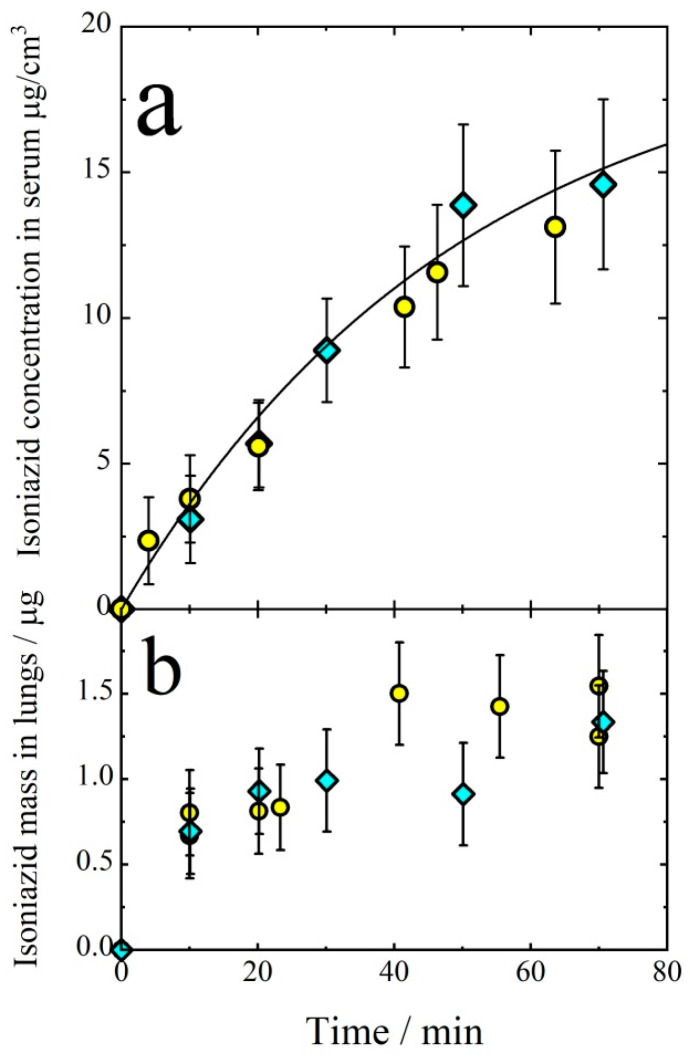
Isoniazid concentration in the blood serum (**a**) and its mass in lungs (**b**) as a function of inhalation time. Diamonds and circles refer to the expositions in WB and NO inhalation chambers, respectively. Solid line follows Equation (11). The rate of aerosol delivery is *I* = 10 μg/min.

**Figure 3 antibiotics-11-01527-f003:**
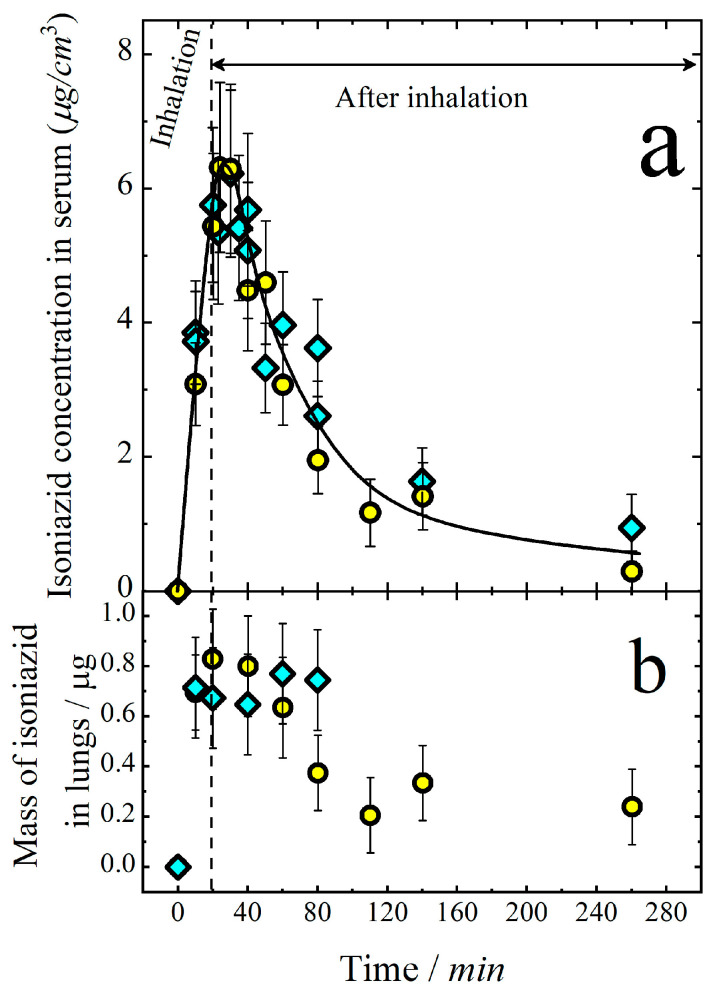
Isoniazid concentration in serum (**a**) and its mass in lungs (**b**) vs. time during and after inhalation. Solid line is an eye guide. Vertical dash line marks the termination of inhalation at time 20 min. The inhalation dose at the end of exposition is 9 mg/kg. Diamonds and circles refer to the expositions in WB and NO inhalation chambers, respectively.

**Figure 4 antibiotics-11-01527-f004:**
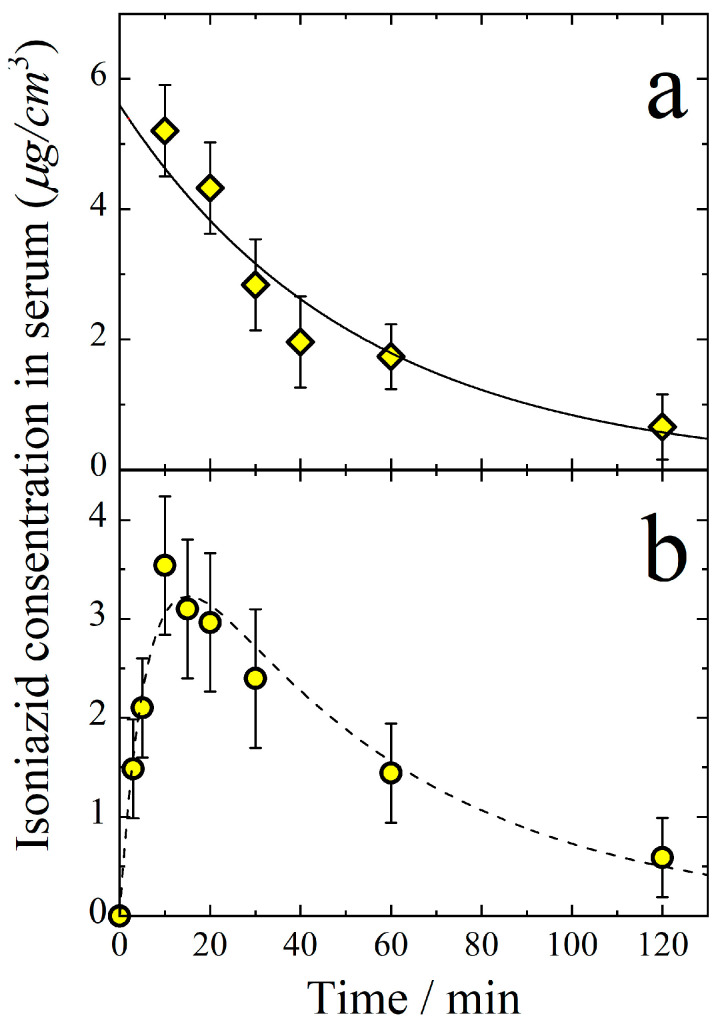
Temporal dependence of isoniazid concentration in the blood serum after intravenous (**a**) and per-oral (**b**) administration. The body-delivered dose is 5 mg/kg. Solid line follows the isoniazid concentration decay function *C* = exp(−*k_e_t*) with *k_e_* = 0.019 min^−1^. Dash line follows Equation (18).

**Figure 5 antibiotics-11-01527-f005:**
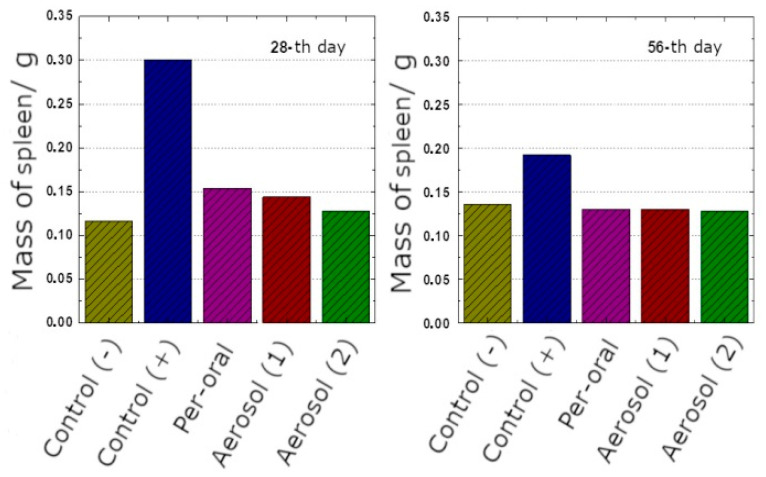
Spleen mass in the animals of experimental groups on the 28th and 56th days.

**Figure 6 antibiotics-11-01527-f006:**
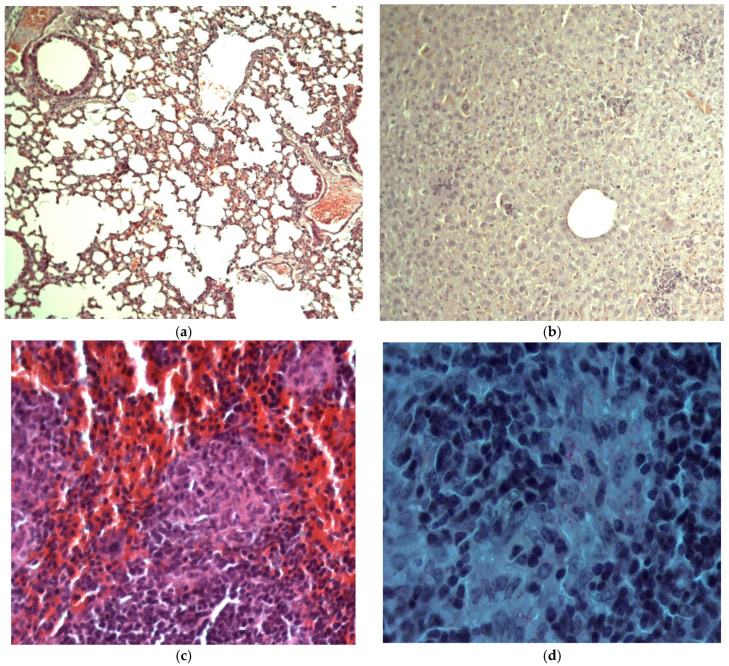
Histologic preparations of lungs (**a**), liver (**b**), and spleen (**c**,**d**) of the animals from the Control (+) group, coloration with hematoxylin-eosine, magnification 20×, MBT in granuloma—Ziehl-Neelsen coloration, magnification 40×.

**Table 1 antibiotics-11-01527-t001:** Experimental groups of laboratory mice.

Group No.	Group Title	Experimental Action
1	Control (−)	Not infected. Inhalation with pure air for 20 min in the whole-body chamber, once a day.
2	Control (+)	Infected. Inhalation with pure air for 20 min in the whole-body chamber, once a day.
3	Per-oral	Infected. Treated with isoniazid, administered through gastric tube in the form of aqueous suspension once a day in the dose of 10 mg/kg.
4	Aerosol (1)	Infected. Treated with isoniazid through inhalation for 20 min in the whole-body chamber once a day. Dose 5.0 ± 0.5 mg/kg.
5	Aerosol (2)	Infected. Treated with isoniazid through inhalation for 20 min in the whole-body chamber once a day. Dose 8.0 ± 0.8 mg/kg.

**Table 2 antibiotics-11-01527-t002:** Number of granulomas and MBT per visual field in the spleen on the 28th and 56th days after infection (M ± SD), *—*p* < 0.05 with respect to the Control (−) group.

Group	28th Day		56th Day	
	Number of Granulomas	Number of MBT	Number of Granulomas	Number of MBT
Control (+)	0.30 ± 0.89	1.30 ± 12.05	0.23 ± 0.54	0.65 ± 2.89
Per-oral	0.03 ± 0.19 *	0 *	0.01 ± 0.09 *	0 *
Aerosol (1)	0.01 ± 0.08 *	0 *	0.02 ± 0.17 *	0 *
Aerosol (2)	0.01 ± 0.11 *	0 *	0.01 ± 0.09 *	0 *

**Table 3 antibiotics-11-01527-t003:** Average number of AFM per 100 visual fields and CFU in the plates after the inoculation of the organs of experimental animals, *—*p* < 0.05 with respect to Control (+) group.

Group	AFM		CFU	
	28th Day	56th Day	28th Day	56th Day
Control (+)	105.5 ± 57.7	67.6 ± 46.5	77.2 ± 67.3	22.6 ± 18.6
Per-oral	0.17 ± 0.23 *	0 *	0.5 ± 0.76	0
Aerosol (1)	0 *	0 *	0	0
Aerosol (2)	0 *	0 *	0	0

## Data Availability

Experimental data are presented in the text.

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
