# Peer review of "Studies of the Specific Activity of Aerosolized Isoniazid against Tuberculosis in a Mouse Model"

_antibiotics, 2022, doi:10.3390/antibiotics11111527_

Round 1

Reviewer 1 Report

General comments

The manuscript required considerable language editing to allow clarity for the reader.

The references should also be formatted according to the journal's requirements (This is just technical).

The authors studied the specific effect of aerosolized isoniazid on tuberculosis using a mouse model. This should reflect clearly in the title. As it is, it appears as if tuberculosis was the model. I suggest modifying the title to "Studies of the specific activity of aerosolized isoniazid against tuberculosis in a mice model,"

The abstract should be improved. The guidelines allow up to 200 words, and the authors should use this. The authors conclude the abstract in a manner that leaves the reader with the question, "and so what?". The authors need to include an implication or a recommendation statement at the end of their abstract. The authors must also include some histology results in the abstract

In the introduction, the authors make key statements without providing references. This must be corrected. Please cite appropriately. For example, "The treatment of drug-resistant TB is especially lengthy (up to two years) and involves the second-line drugs that frequently cause severe side effects and complications. In addition, some new anti-tuberculous drugs exhibit low bioavailability for per-oral administration, for example, due to poor solubility in water, so new routes of administration are to be elaborated."

The Results and Discussion section is grossly lacking. The authors present results but fail to discuss their observations. This must be addressed properly.

Author Response

The manuscript required considerable language editing to allow clarity for the reader.

- We have made additional language editing and hope that now the manuscript is better readable.

 The references should also be formatted according to the journal's requirements (This is just technical).

- The actual  journal requirements say: “Your references may be in any style, provided that you use the consistent formatting throughout. It is essential to include author(s) name(s), journal or book title, year of publication, volume and issue, and pagination”. We have checked that all the necessary parameters in our reference list follow in the correct order. If any other special style is to be followed, we will make amendments as soon as we are aware of the necessary style.

The authors studied the specific effect of aerosolized isoniazid on tuberculosis using a mouse model. This should reflect clearly in the title. As it is, it appears as if tuberculosis was the model. I suggest modifying the title to "Studies of the specific activity of aerosolized isoniazid against tuberculosis in a mice model,"

- We have corrected the title of our manuscript in the revised version.

 The abstract should be improved. The guidelines allow up to 200 words, and the authors should use this. The authors conclude the abstract in a manner that leaves the reader with the question, "and so what?". The authors need to include an implication or a recommendation statement at the end of their abstract. The authors must also include some histology results in the abstract

- We have improved the Abstract. Now the final fragment is: Histologic examination shows that only few macrophagal (nonspecific) granulomas without mycobacteria are detected in the spleen after per-oral and aerosol treatment, the number of granulomas on the 28th day being 3 times smaller in the latter case. Results show that the developed technique of isoniazid aerosol inhalation may have a clinical potential.  

In the introduction, the authors make key statements without providing references. This must be corrected. Please cite appropriately. For example, "The treatment of drug-resistant TB is especially lengthy (up to two years) and involves the second-line drugs that frequently cause severe side effects and complications. In addition, some new anti-tuberculous drugs exhibit low bioavailability for per-oral administration, for example, due to poor solubility in water, so new routes of administration are to be elaborated."

 - The references have been added in the revised version of our manuscript.

The Results and Discussion section is grossly lacking. The authors present results but fail to discuss their observations. This must be addressed properly.

- We edited this section to fill in the lacking information, to make comparison with the efficiency of intravenous and per-oral delivery more informative.

Reviewer 2 Report

Valiulin et al. focused on investigating the efficacy of inhaled isoniazid in a mouse model infected by the the Mtb reference strain H37Rv. Aerosol inhalation of anti-TB drugs can lead to improved efficacy, reduced doses, and elimination of side effects. The results showed the same effectiveness of inhaled INH at a dose of 5 mg/kg compared to orally administered INH at a dose of 10 mg/kg. The study has a clinical potential, however, I have some comments that will need to be clarified. Based on the comments below, I encourage authors to make a few edits before publishing:

Comments:

In introduction section – there is an information: third of the whole population are infected. Please, update these information: third of the entire population is infected with latent TB infection.

In introduction section - I strongly encourage authors to introduce the reader to the issue, so you should add the following paragraph after the sentence „Ten million people become new patients every year, about 450 thousand of them suffer multidrug-resistant TB, and about four thousand people die every day from TB“: In order to reduce the development and spread of resistant forms of TB, rapid diagnosis (for example by whole-genome sequencing) and the correct setting of the treatment regimen, including the appropriate route of administration, are essential (references: https://doi.org/10.1016/j.jgar.2020.02.029; https://doi.org/10.1038/s41598-022-11287-5; WHO - 978-92-4-004257-5).

Have you observed any side effects in mice? If so, were there any specific differences between the Per-oral, Aerosol 1 and Aerosol 2 groups? Its important to mention it in the article.

In Figure 3. concentration of INH after expositions in WB chamber in lungs is rising also after inhalation. May you explain this phenomenon?

Have you considered testing the effectiveness of INH concentrations lower than 5 mg/kg?

In Figure 5. mass of spleen dropped significantly in the untreated group and, on the contrary, increased in the control group. May you explain this?

Did you compare the pharmacokinetics of aerosolized vs orally administered isoniazid?

Per-orally administered isoniazid may be given in a single dose/daily or higher dose/72h. Is it possible to give the patients the higher concentration of isoniazid by inhalation without the need of daily dosing?

Author Response

In introduction section – there is an information: third of the whole population are infected. Please, update these information: third of the entire population is infected with latent TB infection.

- We updated the information according to this recommendation.

In introduction section - I strongly encourage authors to introduce the reader to the issue, so you should add the following paragraph after the sentence „Ten million people become new patients every year, about 450 thousand of them suffer multidrug-resistant TB, and about four thousand people die every day from TB“: In order to reduce the development and spread of resistant forms of TB, rapid diagnosis (for example by whole-genome sequencing) and the correct setting of the treatment regimen, including the appropriate route of administration, are essential (references: https://doi.org/10.1016/j.jgar.2020.02.029https://doi.org/10.1038/s41598-022-11287-5; WHO - 978-92-4-004257-5).

- We are grateful for these references. We have included them into Introduction.

Have you observed any side effects in mice? If so, were there any specific differences between the Per-oral, Aerosol 1 and Aerosol 2 groups? Its important to mention it in the article.

- No obvious signs of side effects were detected. By the end of experiment, there were no visible differences between the Control-, Per-oral, Aerosol 1 and Aerosol 2 groups. The absence of side effect observations may be due to rather short time of experiment.

In Figure 3. concentration of INH after expositions in WB chamber in lungs is rising also after inhalation. May you explain this phenomenon?

- We added the following explanation to this phenomenon: "The concentration of isoniazid in serum and its mass in lungs during and after inhalation is shown in Fig. 3. Both the concentration of isoniazid in blood and its mass in lungs increase with time during the inhalation. After the termination of inhalation, the mass of isoniazid in lungs starts to decrease due to the absorption to blood circulation. However, the drug concentration in serum starts to decrease only about 20 minutes after the termination of inhalation. This delay in serum concentration decrease is because the mass of isoniazid in the respiratory system is still high at inhalation stop time and lung-to-blood absorption is still concurrent with the drug elimination from blood."

Have you considered testing the effectiveness of INH concentrations lower than 5 mg/kg?

- No, lower concentrations were not tested at this stage because therapeutic dosage recommendations are usually 10-15 mg/kg. Even 5 mg/kg is a low value from this point of view, so, to minimize the number of animals used per experiment, we did not test lower doses.

In Figure 5. mass of spleen dropped significantly in the untreated group and, on the contrary, increased in the control group. May you explain this?

- There were two control groups in our experiment: Control- (not infected, not treated) and Control+ (infected, not treated). Mass of spleen increased significantly in Control+ group, that is, in the infected mice that were not treated with isoniazid, just inhaled pure air. So, mass of spleen did not exhibit any unexpected variations.   

Did you compare the pharmacokinetics of aerosolized vs orally administered isoniazid?

- Yes, and in the revised version of our manuscript we added Figure 4b illustrating pharmacokinetics of orally administered isoniazid and added the following discussion: "It is of interest to compare the temporal dependence of isoniazid concentration in serum resulting from per-oral delivery (Fig. 4b) with that for IV and aerosol ways of administration. The rate of absorption from the gastrointestinal (GI) region to blood (WGI) can be approximated by the first-order kinetics

                                                    (16)

where kG is the first-order rate constant for isoniazid transfer through the GI barrier, CGI is the equivalent mass concentration of isoniazid in GI tract, t (min) is time. Then the kinetic equation for isoniazid concentration C(t) in serum can be written as

                                                          (17)

The joint solution of Eqs. (16, 17) is

                                                         (18)

where  is the initial equivalent mass concentration of isoniazid in GI region. To fit the experimental points presented in Fig. (4b) by Eq. (18), three parameters kGI, ke and  are to be adjusted. The best fit values are kGI = 0.16±0.005 min-1, ke = 0.019±0.005 min-1 and  = 4.3±4 mg/cm3. It is important to note that the quantity ke determined from the data on the per-oral administration is in good agreement with that determined from the IV delivery (Fig. 4a). The volume of distribution determined from the per-oral delivery experiments is

                                   (19)

where DGI = 5 mg/kg = 110 mg is the per-oral dose. The volume of distribution determined for GI administration is in good agreement with that obtained from both aerosol and IV delivery experiments."

Per-orally administered isoniazid may be given in a single dose/daily or higher dose/72h. Is it possible to give the patients the higher concentration of isoniazid by inhalation without the need of daily dosing?

- Further studies are needed to give a well-reasoned answer for the case of humans. Results of pharmacokinetic studies with the mouse model suggest that it may be possible to deliver higher isoniazid doses than those delivered within our experiments. However, for mice, we detected that the mass of isoniazid in lungs came to saturation after inhalation for about 30-40 min, as shown in Fig. 2b. This fact may mean that the rate of isoniazid elimination from lungs becomes about equal to the rate of lung delivery. It is still unclear how high the upper dose limit may be for humans.  

Round 2

Reviewer 1 Report

Thank you for addressing the comments raised. The added information may still need to be adjusted reagrding the tense as earlier mentioned. This is reported work.